# Microbiome–Metabolome Reveals the Contribution of the Gut–Testis Axis to Sperm Motility in Sheep (*Ovis aries*)

**DOI:** 10.3390/ani13060996

**Published:** 2023-03-09

**Authors:** Mingming Wang, Chunhuan Ren, Penghui Wang, Xiao Cheng, Yale Chen, Yafeng Huang, Jiahong Chen, Zhipeng Sun, Qiangjun Wang, Zijun Zhang

**Affiliations:** 1College of Animal Science and Technology, Anhui Agricultural University, Hefei 230036, China; 2Modern Agricultural Technology Cooperation and Popularization Center of Dingyuan County, Chuzhou 233200, China

**Keywords:** sheep, sperm motility, rumen microorganisms, serum metabolites, seminal plasma metabolites, hormones

## Abstract

**Simple Summary:**

Healthy sperm viability, the core of male fertility, affects the sustainability of livestock breeding. Several studies have shown a strong link between male sperm motility and gut microbial regulation of the host metabolome, testicular function, and gut microbiota. However, few studies have examined the association between the gut microbiome, host metabolome, and testicular function. In this study, the microbiome and metabolome of adult sheep with significantly different sperm viabilities were analyzed. Our results confirm that gut microorganisms and metabolites differ significantly in sheep with different sperm viabilities and are strongly correlated among themselves and with sperm viability. Thus, our study provides new insights into weak spermatozoa in rams.

**Abstract:**

A close association exists among testicular function, gut microbiota regulation, and organismal metabolism. In this study, serum and seminal plasma metabolomes, and the rumen microbiome of sheep with significant differences in sperm viability, were explored. Serum and seminal plasma metabolomes differed significantly between high-motility (HM) and low-motility (LM) groups of sheep, and 39 differential metabolites closely related to sperm motility in sheep were found in seminal plasma metabolomes, while 35 were found in serum samples. A 16S rRNA sequence analysis showed that the relative abundance of HM and LM rumen microorganisms, such as *Ruminococcus* and *Quinella*, was significantly higher in the HM group, whereas genera such as *Rikenellaceae_RC9_gut_group* and *Lactobacillus* were enriched in the mid-LM group. Serum hormone assays revealed that serum follicle-stimulating hormone (FSH) and MT levels were significantly lower in the LM group than in the HM group, whereas serum glucocorticoid (GC) levels were higher in the LM group than in the HM group, and they all affected sperm motility in sheep. *Ruminococcus* and other rumen microorganisms were positively correlated with sperm motility, whereas *Lactobacillus* was negatively correlated with FSH and GCs levels. Our findings suggest that rumen microbial activity can influence the host metabolism and hormone levels associated with fertility in sheep.

## 1. Introduction

The sheep industry has recently gradually developed a large-scale, standardized, and intensive breeding system. However, with the continuous promotion of large-scale breeding, breeding density is too high, seriously limiting the living habits of sheep. In particular, the relatively limited movement area of breeding sheep sheds results in a decrease in the semen quality of the breeding rams and a decrease in the conception rate of ewes, seriously affecting the sustainable development of the sheep industry. Sperm motility and count are key factors for male fertility and successful fertilization. Sperm with low motility is usually associated with incomplete spermatogenesis and defects in sperm malformation that prevents the penetration of the cervical mucus and reaching the site of fertilization [1]. Furthermore, unlike the reproductive tract of cattle, the reproductive tract of sheep and goats prevents a large number of sperm from entering the uterus, requiring more time for sperm to reach the oviduct to fertilize the oocyte [2]. Therefore, having rams with highly viable semen is crucial for sheep breeding.

Numerous studies have shown that gut microbes play a crucial role in spermatogenesis and male fertility [3,4]. Studies have shown a strong link between testicular function and gut microbiota regulation through the host metabolome, particularly the synthesis of metabolites such as secondary bile acids, which are gut derivatives that affect testicular physiology [5]. In addition, beneficial microbiota have been shown to significantly improve spermatogenesis and semen quality in busulfan-impaired spermatozoa [3]. Simultaneously, intestinal imbalances can impair spermatogenesis and reduce semen quality and male fertility [6]. Ding et al. found that a high-fat diet disrupts intestinal flora and leads to an increased abundance of *Bacteroides* and *Prevotella* in the gut of normal mice, along with a decrease in the number of spermatocytes and spermatozoa in the spermatogenic tubules of normal mice, which is associated with a decrease in sperm quality and quantity [6]. Recent studies have shown that an altered abundance of the intestinal *Ruminococcaceae_NK4A214_group* reduces bile acid levels, leading to impaired spermatogenesis and reduced spermatogenic cell numbers [4]. In contrast, dietary fiber supplementation improves the intestinal flora of boars, promotes the production of short-chain fatty acids, and improves spermatogenesis and semen quality [7]. Thus, the correlation between microbiome and semen parameters/fertility is gradually being revealed with the application of 16S ribosomal RNA sequencing to evaluate the microbiome of sterile males. Furthermore, a potential correlation between sperm viability, the microbiome, and organismal metabolite regulation is also being revealed. However, whether ram sperm motility is associated with gut microbial composition and dysfunction requires further exploration.

The intestinal flora may contribute to changes in host metabolism by altering metabolites or by altering digestion and energy absorption, which may interfere with gamete formation. Intestinal flora can metabolize nutrients in the intestine and regulate intestinal metabolites to affect the blood metabolome [8]. When passing through other organs, blood metabolites can affect development or cause diseases [9]. Improved intestinal flora can improve sperm quality by modulating the plasma metabolome and small-intestinal function in mice [10]. Han et al. showed that the potential mechanism by which hydroxytyrosol (HT) improves semen quality promotes intestinal flora to improve plasma metabolites, which consequently promotes spermatogenesis and semen quality [11]. Ding et al. found that high-fat diets disrupted intestinal flora, stimulated host immune responses, induced inflammation, reduced semen quality, and increased the relative abundance of *Bacteroides* and *Prevotella*, which was associated with higher circulating blood endotoxin levels and reduced spermatogenesis [6]. In addition, dysbiosis of intestinal flora can lower bile acid levels, alter vitamin A metabolism, and transfer it to testicular cells through blood circulation, leading to sperm abnormalities [4]. This suggests that intestinal flora can influence sperm function by regulating blood metabolites. In addition, according to Zhao et al., the combined action of the blood and intestinal flora and testicular metabolome under fucoidan oligosaccharide (AOS) treatment attenuates the destruction of spermatogenesis by busulfan [10]. Zhang et al. also reported that fecal microbial transplantation caused changes in testicular lipid metabolism, increasing the levels of testicular metabolites, which are positively correlated with sperm quality, and contributing to the restoration of spermatogenesis [4]. This finding suggests a correlation between testicular metabolites and male fertility. Seminal plasma, as a testicular secretion, is an important medium for sperm survival and function. Seminal plasma metabolites affect downstream and complementary gene and protein expression changes, which may be key regulators of fertility in bulls [12]. Velho et al. found that metabolites such as bull seminal plasma 2-oxoglutarate and fructose play important roles in physiological processes such as sperm motility, energy metabolism, and regulation of metabolic activity and are important biomarkers for assessing fertility. They identified 21 amino acids in the seminal plasma of bulls of different fertility classes, indicating different metabolite patterns in low-fertility bulls compared to high-fertility bulls [13]. Hamamah et al. studied the seminal plasma of fertile and infertile men using nuclear magnetic resonance spectroscopy (NMR) and found significant changes in the concentrations of glyceryl citrate phosphocholine and lactate in patients with azoospermia and oligozoospermia [14]. Meanwhile, 1857 differential metabolites were identified between the high- and low-sperm viability groups in goats. A database of goat seminal plasma metabolomes was established, and some metabolites possibly affecting sperm viability were detected [15]. Another study concluded that seminal plasma from high-fertility bulls contained more tryptamine, taurine, and leucine and less citric acid and isoleucine [16]. Overall, intestinal flora plays an important role in the host’s energy absorption and metabolism, and dysbiosis can lead to metabolic disorders or stimulate inflammatory responses, disrupt endocrine function, and impair the reproductive capacity of the host. However, it is unclear whether high- and low-sperm motility in rams is related to disorders of intestinal microorganisms and altered serum and seminal plasma metabolites.

Therefore, the primary objective of this study was to understand the correlation between high- and low-sperm motility and microbiota by identifying ram gut microbes with high and low-sperm motility using 16S rRNA sequencing. The secondary objective was to explore whether a non-targeted metabolomics-based approach can highlight significant differences between fingerprints obtained from adult ram serum and seminal plasma samples with different sperm viability levels, to identify new reliable markers of fertility, and to elucidate the intestine–testis axis mechanisms affecting spermatogenesis by combining the gut microbiome and serum and seminal plasma metabolome approaches. Characterizing impaired semen quality is promising and may provide new insights into low-motility sperm in rams and human infertility.

## 2. Materials and Methods

### 2.1. Study Subjects

All experimental procedures were approved by the Ethics Committee of the Anhui Agricultural University, Hefei, China (approval number AHAUB2022008). Adult rams with no significant differences in age, weight, height, length, and chest circumference were obtained from Haiqinsheng Eco-Farming Co., Ltd. (Dingyuan, China) (*n* = 16). High-quality silage was supplemented with mixed concentrated feed, and water was provided ad libitum (all test animals were fed under the same conditions). After 3–5 days of abstinence during the breeding season, fresh ejaculates were collected using an artificial vagina (*n* = 2/ram) [17]. Sperm quality parameters assessed by the computer-assisted sperm analysis system (CASAS, Hamilton Thorne, Beverly, MA, USA) according to World Health Organization (WHO) standards [18] included total sperm motility, velocity of curved line (VCL), velocity of straight line (VSL), mean velocity (VAP), vibration index (WOB), and whip frequency (BCF). Sperm motility was divided into the high-motility (HM) group (sperm motility < 50%) and the low-motility (LM) group (sperm motility > 80%) according to the test method described by Yao et al. [19]. The remaining semen samples were separated from spermatozoa and seminal plasma by centrifugation (10,000× *g*, 4 °C, and 10 min). The separated samples were immediately snap-frozen in liquid nitrogen and used for biochemical analyses.

### 2.2. Sample Collection and Preservation

The rumen contents of adult rams were collected through a gastric tube as described by Wang et al. [20]. Briefly, a gastric tube rumen fluid sampler (Colibri Pastoral Technology Co., Ltd., Wuhan, China) is inserted to a depth of about 100 cm so that the probe tip reaches the rumen ventral sac, a large-capacity sterile syringe (100–200 mL) is connected to the rear of the sampling tube, and the large-capacity syringe is pumped to extract a sample of rumen contents. Approximately 10 mL of the initial rumen sample contents were discarded to avoid saliva contamination. The collected surimi was filtered through two layers of sterile gauze and the filtered rumen fluid was collected into sterile 50 mL centrifuge tubes and immediately frozen in liquid nitrogen for DNA extraction and subsequent microbial and metabolite analysis. All test rams were fasted overnight and blood was collected from the jugular vein via venipuncture into Vacuette blood collection tubes. Each blood sample was then centrifuged (4 °C, 3000× *g*, 10 min) to obtain a serum sample, which was then immediately snap-frozen in liquid nitrogen until analysis.

### 2.3. Extraction and Sequencing of Microbial DNA

Nucleic acid extraction was carried out using the TGuide S96 Fecal Genomic DNA Extraction Kit (Beijing Tiangen Biochemical Technology Co., Ltd., Beijing, China). PCR amplification of bacterial full-length 16SrRNA (V1–V9) was performed using primers 27F (5′-AGRGTTTGATYNTGGCTCAG-3′) and 1492R (5′-TASGGHTACCTTGTTASGACTT-3′). PCR products were mixed and purified, and sequencing libraries were generated using the SMRTbell Template Prep Kit (PacBio), and the libraries were tested for concentration (Qubit) and size (Agilent 2100). The final reaction products were purified by AMpure PB Beads and then sequenced on the Sequel II sequencing platform.

### 2.4. Bioinformatics Analysis

Quality filtering, trimming, denoising, and merging of fastq files was performed using the dada2 package [21] in QIIME2 2020.6.0 software [22]. Amplicon sequence variants (ASVs) were obtained, and ASVs with relative abundance less than 0.005% were filtered. Taxonomic annotation of feature sequences was carried out using the Naive Bayes classifier combined with a comparison approach using Silva.138 as a reference database.

### 2.5. Measurement of Serum Biochemical Indicators

Enzyme-linked immunosorbent assay (ELISA) was used to determine the serum follicle-stimulating hormone (FSH), luteinizing hormone (LH), testosterone (T), vitamin D3 (VD3), lipopolysaccharide (LPS), melatonin (MT), and glucocorticoid (GC) levels. The assay kit was purchased from Shanghai Jianglai Biotechnology Co., Ltd. (Shanghai, China), and the assay procedure was carried out in strict accordance with the manufacturer’s instructions.

### 2.6. Metabolomic Sample Preparation and UPLC-Q/TOF-MS/MS Procedures

All serum and seminal plasma samples were stored at −80 °C and thawed on ice prior to analysis. The process involves the following steps: obtain 100 μL of sample (serum, seminal plasma), add 500 μL of extract containing internal standard (methanol acetonitrile volume ratio = 1:1, internal standard concentration 20 mg/L), vortex and mix for 30 s; sonicate for 10 min (ice water bath). Afterwards, all samples were left to stand at −20 °C for one hour. Then all samples were centrifuged (4 °C, 12,000 rpm, 15 min), 500 μL of supernatant was carefully removed in an EP tube, the extract was dried in a vacuum concentrator, 160 μL of extract (acetonitrile water volume ratio: 1:1) was added to the dried metabolites for re-solubilization, vortexed for 30 s, and sonicated in an ice water bath for 10 min. After that, the samples were centrifuged again (4 °C, 12,000 rpm, 15 min). Finally, 120 μL of supernatant was carefully removed from the 2 mL injection bottle, and 10 μL of each sample was mixed into QC samples for machine testing.

The LC system used for metabolomics analysis consisted of Acquity I-Class PLUS UPLC tandem with a Waters Xevo G2-XS QTOF high-resolution mass spectrometer. The column used was a Waters Acquity UPLC HSS T3 column (1.8 µm, 2.1 × 100 mm) purchased for LC separation, and the column temperature was maintained at 25 °C. The flow rate was 0.5 mL/min and the injection volume was 6 μL. For the positive ion mode, mobile phase A contained 0.1% formic acid aqueous solution and mobile phase B contained 0.1% formic acid acetonitrile. For the negative ion mode, mobile phase A contained 0.1% formic acid aqueous solution and mobile phase B contained 0.1% formic acid acetonitrile. For the negative ionization mode, mobile phase A contained 0.1% formic acid aqueous solution and mobile phase B contained 0.1% formic acid acetonitrile. The linear gradient was set as follows: 0–0.2 min: 2%B, 0.2–10 min: 2%B~98%, 10–13 min: 98%B, 13–13.1 min: 98%B~2%B, 13.1–15 min: 2%B.

The Waters Xevo G2-XS QTOF high-resolution mass spectrometer acquires primary and secondary mass spectrometry data in MSe mode under the control of acquisition software (MassLynx V4.2, Waters, MA, USA). Simultaneous dual-channel data acquisition was performed at low-collision energy (2 V) and high-collision energy (10–40 V) with a mass spectrometry scan frequency of 0.2 s a mass spectrometry map. ESI ion source parameters were as follows: capillary voltage at 2000 V (ESI+ mode) and −1500V (ESI− mode), cone voltage (30 V); cone gas flow rate (50 L/ at 120 °C source temperature conditions) h), and desolventizing gas flow rate (800 L/h at 500 °C desolventizing temperature).

### 2.7. Metabolomics Analysis

The raw peak area information was normalized to the total peak area for subsequent analysis. Principal component analysis (PCA) and Spearman correlation analysis were used to determine the reproducibility of samples within the group and quality control samples. The identified compounds were searched in the KEGG, HMDB, and LIPID MAPS databases for taxonomic and pathway information. Based on the grouping information, the multiplicity of differences was calculated and compared, and the *p*-value of significant differences for each compound was calculated using the *t*-test. OPLS-DA modeling was performed using the R language package ropls, and 200 permutation tests were performed to verify the reliability of the model, and the VIP values of the model were calculated using the multiple cross-validation method. The samples were screened for differential metabolites based on the OPLS-DA model, where metabolites with FC > 1.5, *p*-value < 0.05, and VIP > 1 in serum samples were defined as differential metabolites, and similarly, metabolites with FC > 2, *p*-value < 0.05, and VIP > 1 differential metabolites in seminal plasma samples were defined as differential metabolites. Manual data collation, statistical analysis, and metabolite mapping integration were performed using Excel software (Microsoft, version 2019). We then performed pathway analysis of differential metabolites using MetaboAnalyst 5.0 online software (accessed on 2 October 2022, https://www.metaboanalyst.ca/) to integrate all important metabolites into metabolic pathways. All enriched differential metabolites were identified using MetaboAnalyst 5.0 based on the Kyoto Encyclopedia of Genes and Genomes pathway. The hypergeometric test and relative betweenness centrality were used for statistical analysis of the pathways, and the significance of associations between metabolites and typical pathways was based on the ratio of the number of matches to the number of uploaded metabolites and the total number of molecules in the pathway [23]. In subsequent analyses, only pathways with a nominal significance level of *p* < 0.05 were selected [24].

### 2.8. Data Analysis

Differences in non-parametric data between the two groups were analyzed using the Wilcoxon rank sum test. For all other data, the Student’s *t*-test was used to compare the differences between the LM and HM groups. Statistical analyses were performed using SPSS 20.0 software (SPSS, Inc., Chicago, IL, USA). Potential correlations between differential microorganisms, seminal plasma metabolites, serum metabolites, sperm quality parameters, and serum biochemical indicators were collected and analyzed using mixOmics, *p* value < 0.05 was considered statistically significant. Graphs were generated by Prism 9.0 software (GraphPad Software, San Diego, CA, USA).

## 3. Results

### 3.1. Sperm Vitality

Significant differences were observed in sperm motility and sperm kinematic parameters between the two groups (Figure 1, Appendix A). Sperm motility (*p* < 0.001) and concentration (*p* < 0.05) were significantly lower in the low-motility (LM) group than in the high-motility (HM) group, and the HM sperm exhibited a higher velocity (VCL, VSL, and VAP) and velocity index (WOB) than the LM sperm.

### 3.2. Serum Biochemical Index Analysis

The serum biochemical indices results showed that the blood FSH (*p* < 0.05) and melatonin (MT) (*p* < 0.05) levels were significantly lower in the LM group than in the HM group (Figure 2E,F). In addition, serum glucocorticoid (GC) concentration was higher in the LM group than in the HM group (Figure 2G). These results suggest that sheep with significant differences in sperm quality have some differences in serum physiological parameters, implying that these differences are closely related to sheep sperm quality.

### 3.3. Analysis of the Taxonomic Composition of the Bacterial Microbiota

After sequencing eight samples, a total of 119,061 CCS sequences were obtained by barcode identification, yielding at least 13,756 CCS sequences per sample and an average of 14,883 CCS sequences. Subsequently, 48,735 final valid data points were obtained after removing the primers, chimeras, and denoising. The ACE, Simpson, and Shannon diversity indices showed no change in alpha diversity between HM and LM and, briefly, no significant change in species richness and diversity (Figure 3A–C, Appendix A).

Subsequently, we further explored whether there were differences in gut microbial composition between HM and LM and found that the dominant microbial phylum in the rumen microbiota was Firmicutes, followed by Bacteroidetes, Synergistota, Verrucomicrobiota, and Protebacteriota (Figure 4A). The Wilcoxon rank sum test showed (Figure 4B) that Firmicutes, Actinobacteriota, and Protebacteriota abundance was significantly higher in HM at the phylum level, whereas Bacteroidota and Synergistota were significantly higher in LM. The dominant microbial genus in the rumen microbiota was *Prevotella*, followed by *Quinella, Rikenellaceae_RC9_gut_group* and *uncultured_rumen_bacterium* (Figure 4C). In addition, we performed LEfSe (LDA > 2) and Wilcoxon rank sum test analyses at the genus level (Figure 4D; Appendix A), which showed significant enrichment of *Quinella*, *Ruminococcus* and *Christensenellaceae_R_7_group* in the HM group. By contrast, *Prevotella, Anaeroplasma* and *Fretibacterium* was significantly enriched in the LM group. The gut microbial structures of the two groups were further investigated. PCoA analysis (Bray–Curtis) was performed, and the results showed that rumen microorganisms were divided into two different groups: LM and HM (Figure 4E). In addition, the PLS-DA results (R2(Y) = 0.959 and Q2 = 0.888), which also divided the microorganisms into two groups, indicated a significant difference in the microbiota structure between the LM and HM groups.

### 3.4. Multivariate Analysis of Metabolomic Data

Two-dimensional plots of PCA scores according to serum and seminal plasma metabolomics in ion and negative ion modes are shown in Figure 5A,B and Appendix A. The two groups of samples were noticeably separated, indicating significant differences between the LM and HM metabolites. In addition, orthogonal partial least squares discriminant analysis (OPLS-DA) (Figure 5C,D and Appendix A) showed observed results, indicating that serum and seminal plasma metabolism differed between the two groups. From the replacement plots, the Q2-values were all lower than the R2-values, indicating the original model’s validity (Figure 5E,F and Appendix A).

### 3.5. Identification of Serum and Seminal Plasma Metabolites

After UPLC-Q/TOF-MS/MS analysis, a total of 6172 metabolites were obtained from the metabolic profile of seminal plasma. LM and HM shared 5759 species, while 348 and 65 species were unique to LM and HM, respectively (Figure 6A, Appendix A). In addition, 5561 metabolites were obtained from serum samples. The two groups shared 5228, 228, and 105 metabolites specific to LM and HM (Figure 6A, Appendix A). Differential metabolites between these two groups were identified based on variable importance (VIP), fold change (FC), and *p*-value in the projection. We identified 1785 differential metabolites from seminal plasma samples, of which 511 metabolites were upregulated in HM, while 1274 metabolites were significantly enriched in LM (FC = 2, *p* = 0.05, VIP = 1) (Figure 6B, Appendix A). Similarly, 741 differential metabolites were identified in serum samples, 273 were upregulated in HM, and 468 were significantly enriched in LM (FC = 1.5, *p* = 0.05, VIP = 1) (Figure 6C, Appendix A). Between the LM and HM groups, 39 significantly different metabolites, such as eicosatetraenoic acid, catechol, L-histidine, D-lactic acid, D-leucic acid, L-glutamate, raffinose, and pantothenic acid were found in seminal plasma samples (Figure 6D, Appendix A). A total of 35 significantly different metabolites were found in the serum species (Figure 6E, Appendix A), mainly (S,E)-zearalenone, dopamine, eicosopentanoic acid, aciclovir, cAMP, cannabidiol, arachidic acid, and other metabolites.

### 3.6. Identification of Differential Metabolic Pathways in Serum and Seminal Plasma

To further explore the potential metabolic pathways of these metabolites, KEGG enrichment analysis showed that the altered metabolites in seminal plasma were mainly involved in aminoacyl-tRNA biosynthesis (*p* < 0.001), histidine metabolism (*p* < 0.05), pantothenate and CoA biosynthesis (*p* < 0.05), tryptophan metabolism (*p*< 0.05), riboflavin metabolism (*p* < 0.05), nitrogen metabolism (*p* < 0.05), D glutamine and D-glutamate metabolism (*p* < 0.05), and valine, leucine, and isoleucine biosynthesis (*p* < 0.05) (Figure 6F, Appendix A). In addition, the serum metabolites were mainly involved in riboflavin metabolism, glycerophospholipid metabolism, pyrimidine metabolism, and steroid biosynthesis (Figure 6G, Appendix A).

### 3.7. Correlation of Rumen Microorganisms, Serum Metabolome, Seminal Plasma Metabolome, Serum Biochemical Indicators, and Sperm Quality

In this study, potential correlations between differential microorganisms, seminal plasma metabolites, serum metabolites, sperm motility parameters, and serum biochemical indicators were analyzed using mixOmics (Figure 7A and Appendix A). The results showed the highest correlation between the microbiome and serum metabolism group (r = 0.99), between the sperm kinematic parameters and microbiome (*p* = 0.98), and between the sperm kinematic parameters and seminal plasma metabolism group (*p* = 0.98). In addition, the key correlation results of mixOmics were again visualized by Cytoscape in this study (Figure 7B and Appendix A), which revealed a positive correlation between *Ruminococcus* significantly enriched in the HM group and the serum metabolite dopamine. Higher dopamine levels in the HM group were positively correlated with sperm kinematic parameters (motility, VCL, VAP, and VSL) and negatively correlated with physiological indices (GC). In addition, higher enrichment of microorganisms (*Bifidobacterium* and *Roseburia*) in the HM group was positively correlated with seminal plasma metabolites (L-tryptophan L-glutamate, and D-leucic acid), whereas L-tryptophan, L-glutamate, and D-leucine were positively correlated with sperm kinematic parameters. These results imply a close correlation between sperm motility parameters and gut microbes, the serum metabolome, seminal plasma metabolites, and physiological indices in sheep.

## 4. Discussion

Intestinal flora is involved in all aspects of host health, and a causal relationship between intestinal flora and sperm quality has been established in animal models and humans and is highly correlated with the host metabolome [3]. Hormones are the most important environmental factors affecting germ cell development and have a synergistic effect on the homeostasis of testicular metabolism and the progression of spermatogenesis [25]. In this study, we characterized the rumen microbiome, serum, and seminal plasma metabolome of sheep with high-motility (HM) and low-motility (LM) sperm. For the first time, we further revealed the correlation between them and sperm motility in sheep by integrating microbiome, metabolome, and physiological parameters.

Melatonin (MT), a hormone secreted by the pineal gland of the brain, is closely associated with sperm motility. A study found that serum melatonin levels were significantly higher in the normal sperm group than in the low-viability group [26], consistent with the significantly higher results in the HM group than in the LM group in this study. MT is absorbed by the testes through blood circulation and acts on the testicular mesenchyme to prevent apoptosis and restore testicular function [27], directly affecting the function of the testes. In addition, MT has shown great improvements in sperm concentration and viability in patients treated for varicocele, and long-term use of MT has been shown to improve sperm quality and reduce sperm DNA damage [28]. Notably, MT and gut bacteria appear to have a complex functional relationship, and a study found that MT increases the abundance of *Bifidobacterium* [29]. In this study, higher *Bifidobacterium* abundance was also found in the HM group. Intestinal flora can influence MT levels by regulating its essential precursor, L-tryptophan (Trp) [30]. In de-pinealized rats, Trp administration increased serum MT levels and strongly supported MT synthesis in the gastrointestinal tract [31]. In addition, Trp is effective in improving sperm quality, and Trp metabolite fluid in seminal plasma was also significantly higher in normal men than in men with weak spermatozoa [32], consistent with the higher seminal plasma Trp in the HM group than in the LM group in this study. From the above results, the activity of rumen microorganisms (especially *Bifidobacterium*) may directly or indirectly affect sperm motility in sheep by affecting Trp levels or regulating MT levels through Trp.

Dopamine (DA) is an important neurotransmitter that plays a physiological role in regulating viability, fertility, and sperm motility [33]. A study found that in patients with weak sperm and oligospermia, the concentration of DA in the seminal plasma and blood was lower than that in men with normal fertility [34]. In the present study, we obtained similar results as DA in serum metabolites, which was significantly higher in the HM group than in the LM group. DA induces activation of dopamine type 2 receptor (D2DR) in boar sperm, consequently increasing tyrosine phosphorylation and accelerating the movement of sperm [33]. Dopamine improves motility parameters and acrosome responses in highly motile sperm subpopulations (HM), and these effects are thought to be caused by tyrosine phosphorylation [35]. In addition, activated D2DR can lead to elevated Ca^2+^ levels, and the Ca^2+^ content in the middle of the sperm is associated with the flagellum and influences flagellum activity [36]. The above studies provide sufficient evidence for the importance of DA as a neurotransmitter for male reproduction. Therefore, in the present study, serum dopamine metabolism was abnormal in the LM group, leading to reduced sperm motility in sheep. Additionally, Trp levels are closely related to DA anabolism. Depletion of Trp reduces DA biosynthesis in vivo, and the administration of Trp in rats increases DA levels in the striatum of the brain [37]. Notably, microorganisms of genera such as *Prevotella*, *Lactobacillus*, *Bifidobacterium*, and *Rumenococcus* modulate the receptors, transporters, and specific targets of the dopaminergic pathway positively or negatively [38]. They have demonstrated an important link between the central nervous system and dopaminergic pathways in the periphery of the gastrointestinal system [39]. The gut microbiota has been identified as a key regulator of crosstalk between the brain and the gastrointestinal tract (gut–brain axis) [40]. Significant microbiota dysbiosis (significant reduction in *Ruminococcus*, *p* < 0.05) and decreased plasma DA levels in mice were found in CRS (chronic restraint stress) mouse studies, and a positive correlation between *Ruminococcus* abundance and DA was found, suggesting a positive correlation between *Ruminococcus* and DA metabolism [41]. We can speculate that the dysregulation of gut microbes (*Ruminococcu*) in the present study caused abnormal dopamine metabolism in the LM group, leading to decreased sperm motility in sheep. In conclusion, microorganisms can affect DA metabolism and, thus, sperm quality by regulating DA metabolism or Trp levels.

In addition, in this study, serum Glucocorticoids (GC) was higher in the LM group than in the HM group. Excessive levels of an effective anti-inflammatory hormone in the body may impair male reproductive function. A study revealed that dexamethasone (DEXA) affects the gonadotropin axis, inhibits testosterone production, adversely affects testicular tissue, and affects testes and spermatogenesis by reducing daily sperm production and disrupting sperm viability [42]. In addition, DEXA can induce excessive production of reactive oxygen species (ROS) and cause oxidative stress, leading to impaired sperm parameters, testicular damage, and ultimately, male infertility [43]. Excessive activation of the hypothalamic–pituitary–adrenal axis (HPA) owing to high GC levels is the main cause of impaired gonadal function and fertility [44]. GC induces apoptosis in mouse spermatogenic cells. The rapid and efficient degradation of apoptotic germ cells by Sertoli cells is essential for developing and differentiating germ cells and is a necessary process for spermatogenesis to proceed in healthy germ cells [44]. If the ability of Sertoli cells to phagocytose is impaired, it leads to an increase in the number of apoptotic germ cells, which cannot be eliminated and converted into energy [45], resulting in a non-infectious inflammatory response in the testis. In addition, Sertoli cells provide morphological support through cell–cell interactions and biochemical components through the secretion of lactic acid [46]. Lactic acid, derived from glucose metabolism by supporting cells, is the main metabolic fuel for post-meiotic germ cells and inhibits apoptosis of testicular germ cells [47]; it is a good energy substrate for sperm survival and motility. In the present study, higher concentrations of D-lactic acid were found in the seminal plasma of the HM group. However, GC can cause a decrease in lactate content in testes and TM4 cells and a downregulation of phagocytic activity in Sertoli cells, accompanied by a decrease in mitochondrial activity through the upregulation of PDK4 (Recombinant Pyruvate Dehydrogenase Kinase 4) [48]. In addition, elevated serum CORT (Cortisol) levels induce p27 (a cell cycle protein-dependent kinase inhibitor) expression in Sertoli cells and terminate Sertoli cell proliferation, leading to a decrease in the number of Sertoli cells in mouse testes [49]. From the above results, GC can affect the value addition of Sertoli cells and induce disturbance of lactate metabolism in Sertoli cells, affecting their phagocytic ability, causing testicular inflammation, and impairing reproductive function. Notably, a correlation existed between GC and the activity of the gut microbes. Gut microbiota regulates glucocorticoid levels [50]. Plasma ACTH (Adrenocorticotropic Hormone) and corticosterone levels were reduced in *Bifidobacterium infantis* mono-associated mice, whereas plasma ACTH and corticosterone levels were increased in *E. coli* mono-associated mice [51]. *Lactobacillus* was found to enhance the potentiation of prednisone in EAH(experimental autoimmune hepatitis) mice [50]. In addition, serum ACTH and corticosterone levels were higher in germ-free mice after acute stress than in specific pathogen-free mice [51]. These studies suggest a close association between microorganisms and glucocorticoids. In the present study, *Lactobacillus* was detected in the rumen fluid of the LM group but not in the HM group. Therefore, we can hypothesize that the gut microbiota (*Lactobacillus*) can influence the host behavior and HPA(Hypothalamic-pituitary-adrenal) axis secretion of GC, causing a decrease in the number of Sertoli cells and disturbance of lactate metabolism, consequently impairing reproductive function and affecting sperm quality in sheep. However, this study did not validate the proliferation status and metabolic processes of Sertoli cells; therefore, further studies are required.

A study showed that follicle-stimulating hormone (FSH) in the serum of oligozoospermic and hypospermic males was significantly lower than that in normal males [52], consistent with the results of the present study. In contrast, FSH treatment significantly improved routine sperm parameters [53]. This suggests that applying FSH when treating patients with oligo- and hypospermia can be an effective treatment strategy. In addition, probiotics can affect the host’s reproductive function by regulating the host metabolism. Probiotic (*Lactobacillus rhamnosus*) supplementation increases serum FSH levels in males and results in increased sperm velocity (VSL, VCL, and VAP) and the percentage of progressively motile sperm [54]. Wen et al. found a negative correlation between the relative abundance of *Lactobacillus* and serum FSH levels (r = −0.27, *p* = 0.046) [55]. In the present study, we detected *Bifidobacterium* in the HM midgut, which was not in the LM group. In addition, *Lactobacillus* was found in the LM group, whereas it was not detected in the HM group. Thus, gut microbes (*Bifidobacterium* and *Lactobacillus*) contribute to the differences in sperm quality in sheep by regulating host FSH levels. This process possibly results from gut microbes affecting testicular development and function by regulating the permeability of the blood–testis barrier and modulating serum FSH levels [56]. However, this process requires further investigation.

Microorganisms can also regulate sperm viability by affecting host amino acid metabolism. Most of the seminal plasma metabolome differences in this study focused on amino acid metabolic pathways, suggesting the importance of amino acid metabolism in maintaining sperm viability. A study demonstrated the important role of amino acid metabolic pathways in the regulation of parameters related to semen quality [57]. Amino acid metabolism disorders are thought to be associated with structural and functional alterations in the spermatozoa of men with severe oligospermia. Zhao et al. found reduced levels of several amino acids, such as leucine (Leu), glutamic acid (Glu), and Trp, in patients with weak spermatozoa [58]. Another study found lower levels of tryptophan and glutamate in men with lower sperm motility and higher rates of morphological abnormalities [59]. In the present study, the levels of Trp (FC = 2.29, *p* < 0.001), Glu (FC = 2.88, *p* < 0.001), and Leu (FC = 2.95, *p* < 0.001) in the seminal plasma were also significantly lower in the LM group than in the HM group. Trp leads to increased testicular descent and reduced spermatozoa, whereas Trp-supplemented diets can significantly improve sperm motility in rams [60]. In addition, the addition of amino acids to the diet can improve sperm quality, change seminal plasma composition, and improve sperm fertility in boars, and these effects are related to Ca^2+^ and cAMP synthesis [61]. All of the above studies showed the importance of amino acids in regulating sperm quality. Changes in amino acid composition also play an important role in altering the gut microbiota. Leu has been reported to promote intestinal development in piglets [62]. Glu significantly alters the composition of the intestinal microbial community, increases the diversity of the microbial community, and promotes the colonization of *Roseburia* [63]. Metabolic changes associated with the rat cecum microbiome indicated that L-glutamate was negatively correlated with *Prevotella* [64]. Although amino acids can alter gut microbiota, they can also maintain host amino acid homeostasis by facilitating amino acid digestion and absorption. For example, the pig gut microbiota promotes the synthesis of essential amino acids such as Leu, which the host requires [65]. These studies show that amino acids can regulate the gut microbiota, which can also influence the metabolic processes of amino acids, and that their interaction can affect male fertility. Therefore, we hypothesized that in the present study, differences in sperm quality between LM and HM groups of sheep were associated with the dysregulation of amino acid metabolism caused by disorders of the genera *Prevotella*, *Bifidobacterium*, and *Roseburia*.

## 5. Conclusions

In conclusion, we provide a unique perspective for studying the correlation between the intestine–testis axis in sheep. Rumen microbial activity can influence sperm motility in sheep by affecting Trp metabolism and regulating serum MT and DA levels. In addition, elevated glucocorticoid levels owing to rumen microbial disorders can cause disturbances in lactate metabolism, impair reproductive function, and affect sperm motility in sheep. FSH levels, which are closely related to sperm motility, are modulated by microorganisms. There were also differences in the amino acid metabolism levels in the seminal plasma of LM and HM rams. This difference was closely related to rumen microbial activity, suggesting that sperm quality in sheep is related to the level of amino acid metabolism regulated by microorganisms.

## Figures and Tables

**Figure 1 animals-13-00996-f001:**
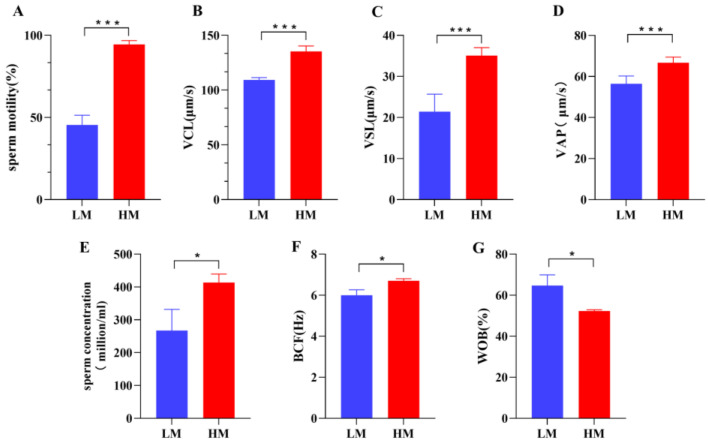
There were significant differences in various kinematic parameters of sperm between the LM and HM groups: (**A**) sperm motility (%), (**B**) VCL (curvilinear velocity, μm/s), (**C**) VSL (linear velocity, μm/s), (**D**) VAP (average path velocity, μm/s), (**E**) sperm concentration (million/mL), (**F**) BCF (beat/cross frequency, Hz), (**G**) WOB (wobble, %). * *p* < 0.05, *** *p* < 0.001. LM, low-motility group; HM, high-motility group.

**Figure 2 animals-13-00996-f002:**
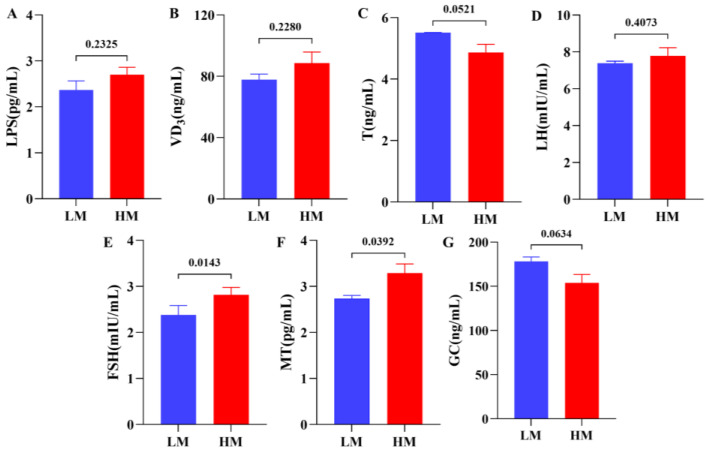
Serum physiological parameters between LM and HM groups: (**A**) LPS (lipopolysaccharides, pg/mL), (**B**) VD_3_ (vitamin D3, ng/mL), (**C**) T (testosterone, ng/mL), (**D**) LH (luteinizing hormone, mIU/mL), (**E**) FSH (follicle-stimulating hormone, mIU/mL), (**F**) MT (melatonin, pg/mL), (**G**) GC (glucocorticoid, ng/mL). LM, low-motility group; HM, high-motility group.

**Figure 3 animals-13-00996-f003:**
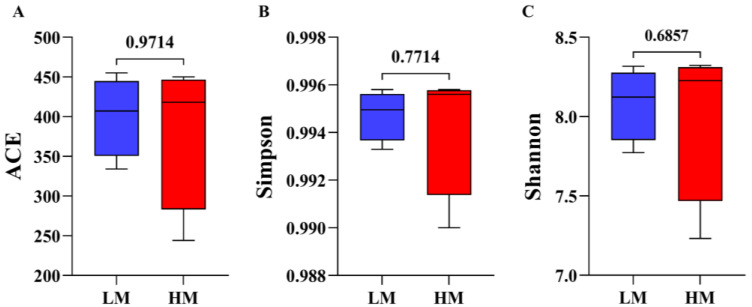
Alpha-diversity analysis of gut microbial community structure in LM and HM groups (**A**) ACE index: (**B**) Simpson index, (**C**) Shannon index. Red represents the HM group; blue represents the LM group. LM, low-motility group; HM, high-motility group.

**Figure 4 animals-13-00996-f004:**
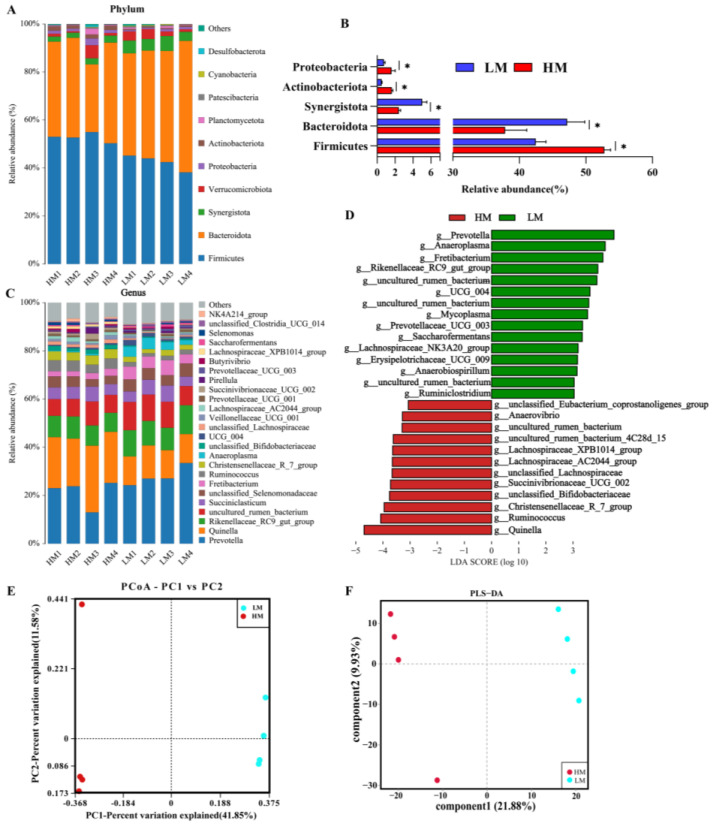
Rumen microbial composition and diversity between LM and HM groups of sheep. (**A**) Taxonomic composition of rumen microorganisms at the phylum level between the two groups. (**B**) Significantly differentiated microbes at the phylum level (* *p* < 0.05, determined by Wilcoxon rank-sum test). (**C**) Taxonomic composition of rumen microorganisms at the genus level between the two groups (1%, according to relative abundance). (**D**) LEfSe analysis at the genus level (LDA > 2). (**E**) PCoA analyses based on Bray−Curtis distance. (**F**) Analysis of the ruminal microbiota using PLS-DA. LM, low-motility group; HM, high-motility group.

**Figure 5 animals-13-00996-f005:**
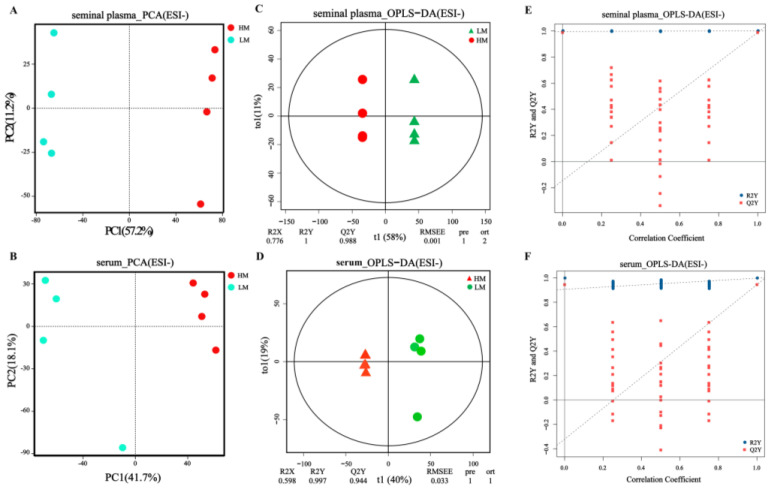
Rumen microbial composition and diversity between LM and HM groups of sheep. (**A**) Taxonomic composition of rumen microorganisms at the phylum level between the two groups. (**B**) Significantly differentiated microbes at the phylum level (*p* < 0.05, determined by Wilcoxon rank-sum test). (**C**) Taxonomic composition of rumen microorganisms at the genus level between the two groups (1%, according to relative abundance). (**D**) LEfSe analysis at the genus level (LDA > 2). (**E**) PCoA analyses based on Bray−Curtis distance. (**F**) Analysis of the ruminal microbiota using PLS−DA. LM, low-motility group; HM, high-motility group.

**Figure 6 animals-13-00996-f006:**
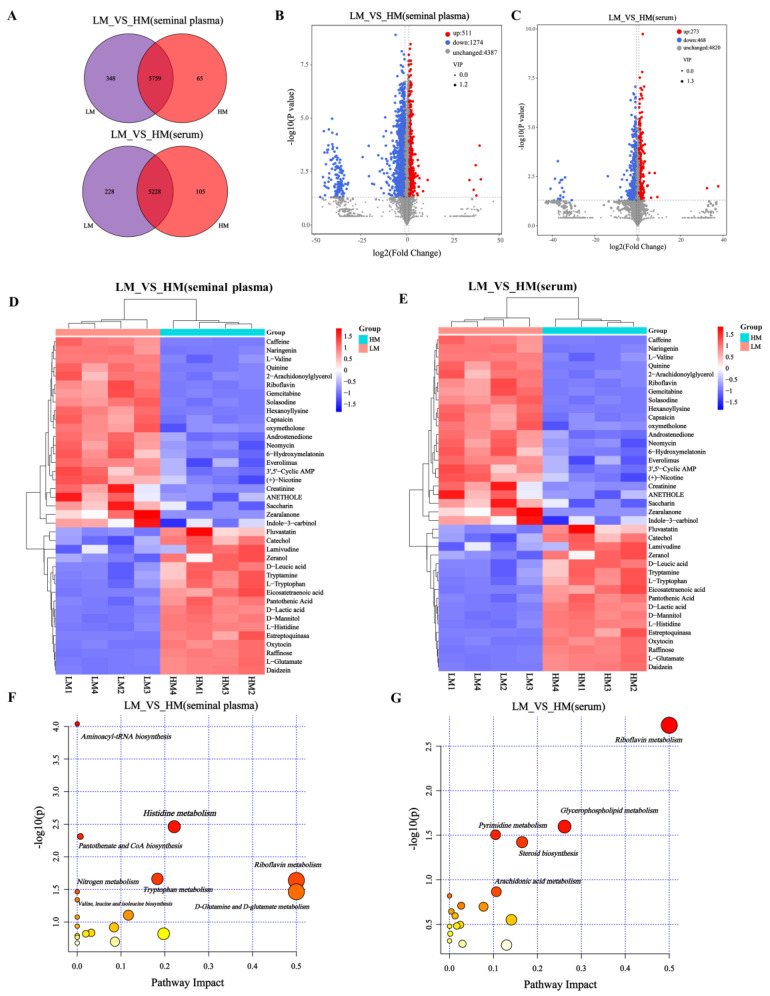
Identification of serum and seminal plasma metabolites and analysis of metabolic pathways. (**A**) Venn diagram of seminal plasma and serum metabolites in LM and HM groups. (**B**,**C**) Volcano maps of metabolites in different groups of seminal plasma (FC = 2, *p* = 0.05, VIP = 1) and serum (FC = 1.5, *p* = 0.05, VIP = 1). (**D**,**E**) Heat maps of serum and seminal plasma differential metabolites between LM and HM groups. (**F**,**G**) Disturbed metabolic pathways in serum and seminal plasma samples of LM and HM groups were performed by MetaboAnalyst5.0. LM, low-motility group; HM, high-motility group.

**Figure 7 animals-13-00996-f007:**
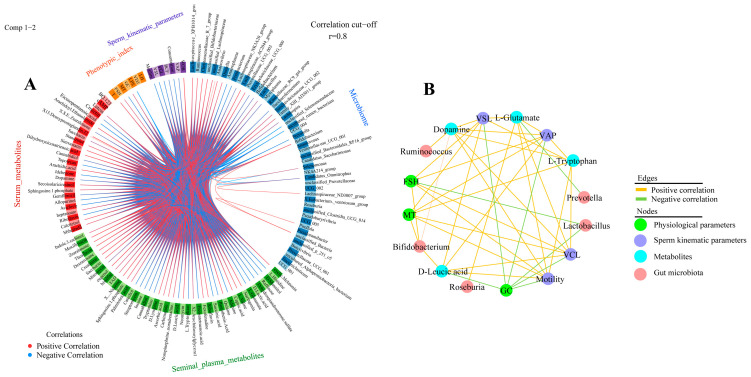
Microbiome and metabolome correlation analysis (**A**) Circos plot showing correlations between microbiome, serum metabolome, seminal plasma metabolome group, physiological parameters, and sperm kinematic parameters, with positive correlations indicated by red lines and negative correlations by blue lines. Only results for R > 0.8 are shown. (**B**) Based on DIABLO analysis results, Cytoscape shows correlations between key microbial genera, metabolites, physiological parameters, and sperm motility parameters.

## Data Availability

All data generated and analyzed during this study are included in this published article. Raw data supporting the findings of this study are available from the corresponding author on request.

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
