# Peer review of "Microbiome–Metabolome Reveals the Contribution of the Gut–Testis Axis to Sperm Motility in Sheep (Ovis aries)"

_animals, 2023, doi:10.3390/ani13060996_

Round 1
Reviewer 1 Report
Animals-2238989.
Microbiome-metabolome reveals the contribution of the gut- 2 testis axis to sperm motility in sheep (Ovis aries).
This is an interesting investigation about sheep reproduction, which aims to describe the characteristics of the sperm and the relationship with the bacterial communities from gut microbiota.
The objective of the work is to assess association among the testicular function, bacterial gut microbiota, and the metabolism of the host. Thus, it describes the differences between serum and seminal plasma metabolomes and the motility of spermatozoa.
The work was supported by a robust methodology to determine sperm parameters, serum and seminal plasma metabolites, identification of differential metabolic pathways, metagenomic from the rumen microbiota; also, a powerful data analysis was performed to detect significant differences between the sperm viability and other parameters.
My major concern is the low number of samples included in the experimental design: animals=16, ejaculates=32, rumen contents=16, blood samples=16; however, the results shows only n=4/group (total=8). This review considers that it represents a limitation in the interpretation of the results. Perhaps the authors could discuss this issue.
Some corrections are necessary before publishing:
Line 32: always explain acronyms the first time you use one.
Line 72: ... the application of 16S ribosomal RNA sequencing to the microbiome of sterile males” replace by: “the application of 16S ribosomal RNA sequencing to evaluate the microbiome of sterile males.”
Lines112-113: “Another study concluded that seminal plasma from high-fertility cows contained more tryptamin…” replace by “Another study concluded that seminal plasma from high-fertility bulls contained more tryptamin…”
Lines 125-126: delete “, for the first time,”
Line 145: “… described by Yao et al. [19] test method.” Replace by: “….described by Yao et al. [19]”.
Line 146: check format “4 °C”.
Line 151 and 205: add space.
Figure 7: “Shysiological” replace by “Physiological”
Line 275 and 393: “flora” replace by “microbiota”.
Line 397: “endostasis” replace by “homeostasis”.
Line 402: Melatonin.
Line 422: always explain acronyms the first time.
Line 426: sentence must be moved to results section: “…, which was significantly higher in the HM group than in the LM group (FC=1.56, p<0.001)”.
Line 474: always explain acronyms the first time.
Line 476: always explain acronyms the first time.
Line 477: “p27 protein”.
Line 480: “ ….., affecting the phagocytic ability of Sertoli cells,..” replace by : “affecting their phagocytic ability,..”.
Line 483: always explain acronyms the first time.
Line 484: E. coli (italic).
Line 547-549: “Rumen microbial activity can influence sperm motility in sheep by affecting Trp metabolism levels and regulating serum MT and DA metabolism levels” replace by “Rumen microbial activity can influence sperm motility in sheep by affecting Trp metabolism and regulating serum MT and DA levels”.
Line 552: “metabolism levels”?
On basis of these observations, I consider that it is a good work; however, the manuscript must be revised.
Author Response
请参阅附件

Reviewer 2 Report
This is a high-quality study with methods consistent with the objectives set, with the generation of a large amount of information of interest, a deep analysis and a very good focus on the discussion and conclusions. It is recommended to review the following aspects:
- Define the abbreviation GC in the abstract
- In section 2.1. Study Subjects, please review the definition of the study groups which seems to be defined in the opposite way "Sperm motility was divided into the high motility (HM) group (sperm motility <50%) and the low motility (LM) group (sperm motility > 80%) according to the test 144 method described by Yao et al. [19] test method"
- What was the criteria used to define the motility groups? The groups do not seem to be distributed homogeneously, and as described, semen samples with motility between 50 and 80% would not be included. Is this so? Could an intermediate motility group have been included?
- In section 2.1. Study Subjects and in the rest of the document, express centrifugation in g-force "The remaining semen samples were separated from spermatozoa and seminal plasma by centrifugation (3000 rpm, 4 °C, and 10 146 min)."
- In the legends of the figures include the complete definition of the evaluated parameters prior to the abbreviations, the figures must be self-explanatory
No additional corrections are required.
Author Response
请参阅附件

Reviewer 3 Report
The current review presents a manuscript which attempts to reveal the correlation between gut-testis axis and sperm motility or quality in sheep. The experiments are presented well and a thorough discussion accompanies the results. In my opinion, there are several areas which the manuscript could benefit from some additional work to help strength the outcomes and overall message of the manuscript.
General comments:
- The introduction could be strengthened by focusing more on WHY the study needs to be completed and the IMPACT or IMPORTANCE this has on sheep breeding in China. For example, on lines 84-88, you mention that high fat diets can disrupt the gut flora and contribute to reduced sperm quality. Are these high fat diets increasing in intensively run systems? Is there a problem with increased gut flora, increased inflammation and sperm quality? Are these common problems that need to be addressed in intensively run programs to improve breeding outcomes of sheep in China?
- A better introduction of the assays to be used and why in the introduction would also be useful to better set up the results and later discussion i.e what is the role of RNA sequencing?
- Overall would refrain from using the term 'weak' as this is very subjective. A cut off of sperm motility would be better.
- I noticed that the concentration of sperm was significantly different between HM and LM groups. This will affect your motility assessment and could likely contribute to a different level of metabolites in the seminal plasma. It would have been better for these treatments to be standardised once divided into HM and LM. Could you please comment on why this wasn't done?
- Several abbreviations were used through out the manuscript which weren't explained appropriately for the reader to understand the text. i.e. Line 265, MT
- In the discussion, it wasn't immediatley clear on the link between changing or variable gut flora composition on variable seminal plasma and sperm quality- what is the immediate cause and effect in the physiology? How does the gut flora impact on the accessory sex gland secertion of vital proteins, sugars and hormones, necessary for sperm survival at ejaculation
- in general, the discussion is well-written with lots of useful evidence to support results, however I would suggest amending the structure of the paragraphs. I would recommend starting with the result from the current student, then addressing whether it was been replicated in different studies or species and then linking it back to the overall message or aim.
- I wouldn't suggest ending your conclusion on future work. This should be included in the last paragraph of the discussion
Specific comments;
- Line 56, indicate which species you are referring too? Where is the gap in the research?
-What was the concentration of sperm run on the CASA?
- Line 253, what is sperm vitality? Please use sperm quality or assessment of sperm parameters
- Line 254, please use the correct term of sperm motility and other kinematic parameters
- Should gut flora microbial species be italicised when written in text? e.g. line 290
- Line 394- what animal species?
- Please expand abbreviations in the discussion
Author Response
请参阅附件
